# Innocell Bioreactor: An Open-Source Development to Produce Biomaterials for Food and Packaging Based on Fermentation Processes



Nitzan Cohen [1,*], Emma Sicher [2], Camilo Ayala-Garcia [1], Ignacio Merino Sanchez-Fayos [1], Lorenza Conterno [3] and Secil Ugur Yavuz [1]

1   Design Friction Lab, Free University of Bozen-Bolzano, 39100 Bolzano, Italy;
    camilo.ayalagarcia@unibz.it (C.A.-G.); ignacio.merinosanchezfayos@unibz.it (I.M.S.-F.);
    secil.uguryavuz@unibz.it (S.U.Y.)
2   Cluster of Excellence Matters of Activity, Image Space Material, Humboldt-Universität zu Berlin,
    10117 Berlin, Germany; emma.sicher@hu-berlin.de
3   Fermentation and Distillation Group Laimburg Research Center, Laimburg 6, 39040 Ora, Italy;
    lorenza.conterno@laimburg.it
*   Correspondence: nitzan.cohen@unibz.it

**Abstract:** A growing number of science and design scholars and design practitioners have recently embarked on studying fermentation processes to produce alternative materials. The main driver of this trend is the search for a sustainable future by proposing novel alternatives that could substitute or integrate into society's current production and consumption models. This study presents the development of an open-source bioreactor capable of enhancing and optimizing a symbiotic culture of bacteria and yeast (SCOBY) production process. The bioreactor is part of a greater design-driven project aiming to process edible and non-edible materials. The study presents the experiments and methods that led to the development and refinement of the current bioreactor, and all the information needed to replicate it with tools and equipment currently available under the Creative Commons status. The aim of sharing open-source methods and results to reproduce the bioreactor is to support different interdisciplinary teams of scientists and designers in generating high amounts of SCOBY, accelerating R&D with this auspicious yet underexplored source of bacterial cellulose.

**Keywords:** microbial cellulose; growing design; fermentation; rotating disk bioreactor; biomaterial production; open source





## 1. Introduction

Since the beginning of human society, materials and their transformation processes have been closely connected to the evolution of humanity as a fundamental process of interaction between humans and the environment [1]. For each material discovered and its consequent possibility of use, humankind has devised diverse methods to efficiently transform matter, skillfully harnessing the resources at their disposal. The advent of industrialization over the last two centuries led to machines capable of transforming matter at important scales, allowing more people to access the different goods created [2]. Since then, the majority of artifacts and foods we interact with originate daily from industrially processed raw resources [3,4]. The recent environmental and social catastrophes we face have placed our entire ecosystem in crisis, depleting non-renewable resources, producing hazardous waste, and perpetuating the inefficient use of energy resources [5,6]. Alternatives are urgently needed to help mitigate the adverse effects of climate change and help produce more conscious and circular models. Part of the creative strategy of design as a discipline lies in observing the different practices existing in other areas and cross-pollination, which allows innovation toward a socioeconomic transformation with a focus on industry for the future [7].

In the analysis of manufacturing and creation processes of the different disciplines, the design practice recognizes materials and their management as critical points for sustainable production and consumption intervention [8] (p. 25). The material, previously considered a step in the design process, now becomes the project's focus [9,10]. Consequently, a material-driven approach becomes an essential tool for sustainability envisioning an effective ecological transition.

Unlike more common engineering and science-driven material developments, material design augments the development of alternatives through speculative scenarios. Instead of focusing on material development to improve performance or a specific market need, active engagement of materials experimentation leads to innovative solutions to contemporary problems.

Several contributions emerged in the international arena in the last ten years, transforming material resources through a process led by design [11]. Some material development alternatives come from understanding the capabilities to grow or harvest a material thanks to agricultural techniques [12], fermentation methods [13], or the manipulation of organic waste. Some other contributions have developed a creative use of parts (hair, skin, bones) of animals [14], or through the collaboration of animals or microorganisms such as bacteria [15]. Others came from the transformation of industrial by-products (stones, metals, polymers, textiles) through alternative methods of recycling or upcycling [16]. Others come from combining materials and technology through printed electronics and smart materials [17].

Many of the materials developed in recent years do not come exclusively from scientific laboratories, as was the standard, but take advantage of the possibilities of interdisciplinary research typical of design. Indeed, experts in multiple fields [18] such as biology, engineering, or agriculture have developed, through a multidisciplinary approach, alternatives to counteract the use and abuse of materials and consumption of products in the current linear path of our industrialized society. The involvement of multiple disciplines and, in some cases, a multicultural approach to materials development, is becoming the new standard [19]. The different and emergent methods available today allow us to imagine alternative and sustainable futures. The conscious use of resources that minimize energy consumption, together with new access to technologies through open source [20], technological democratization [21,22], and the rediscovery of ancient artisanal practices [23], provide the necessary tools to propose different industrialization means or alternative approaches to production [24]. One of the drivers for achieving circularity in industry and agribusiness comes from understanding the value of all the different resources involved in the process [25]. The above-mentioned approach has driven different multidisciplinary teams led by design, including ours, to discover potential elaborations and transformations of new types of materials by taking advantage of the fermentation process commonly used to turn sweetened tea into Kombucha.

Kombucha is believed to have originated in Northeast Asia, possibly in regions such as China or Manchuria, around 200 BC [26]. It subsequently spread to Russia and Europe in the 19th century, gaining popularity for its alleged health benefits, particularly in Germany. Kombucha made its way also to the Americas, primarily through immigrant communities. It gained popularity as an alternative healthy drink, and has gradually increased its popularity globally [27]. With the widespread availability of tea ingredients, starter culture and expertise, fermentation is one of the most available processes that can be used for biomaterials R&D.

Kombucha tea is produced with an infusion of black tea leaves that is sweetened with sugar and in some cases syrups or honey [28]. Green tea or oolong tea leaves may also be used. The fermentation is carried out by a symbiotic culture of bacteria and yeast (known through its acronym SCOBY). The fermentation process conducted for the beverage usually takes between 7 and 14 days, although it can vary depending on the temperature and desired flavor profile. After fermentation, Kombucha is usually strained to remove the SCOBY and any sediment before being bottled and refrigerated for carbonation and later

storage. It is at this moment that this traditional drink attracts the attention of scientists and creatives from different disciplines, who observe the newly generated SCOBY not as a fermentation by-product, but as a promising biomaterial source as it appears as a floating jelly-like layer between the air–liquid interface that is generated by the bacteria present in the culture [29].

This biomaterial appears as a hydrogel layer [30] or, in other words, as a gelatinous, thick, and translucent pellicle [31,32] (Figure 1). The SCOBY contains microbial cellulose (MC) produced by acetic acid (AAB) bacteria, in particular by *Komagataeibacter xylinus* and *Komagataeibacter hansenii*. These bacteria produce an interwoven matrix that embeds the microorganisms and their metabolites, and retains the liquid (SCOBY is composed of over 90% water). MC is secreted outside the bacterial cell in the form of nanofibers [33] that are one hundred times thinner than those of plant-derived cellulose, and whose water retention capacity is one hundred times greater [34]. New biofilms grow on the surface of the liquid because the microbes can more easily come into contact with oxygen, which is essential according to their aerobic nature. The resulting layers of SCOBY possess properties such as high purity, a non-woven nanoscale fiber network, quite uniform morphology, tensile strength, transparency, water-holding capacity, water insolubility, porosity, antimicrobial, chemical stability, non-toxicity, body compatibility, compostability, and permeability to gases and liquids, making it compatible with a large variety of edible and non-edible potential applications [31,32]. The SCOBY layers can be processed into various states ranging from powder to flakes, sheets, gels, or foams [35].

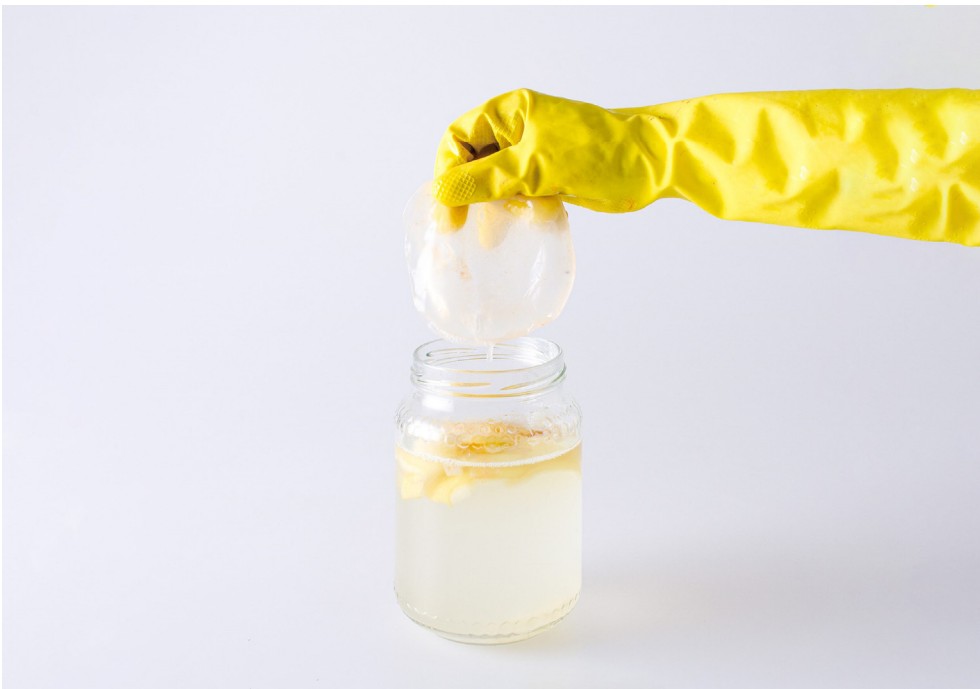

**Figure 1.** SCOBY biofilm after beginning the fermentation cycle.

Its excellent physicochemical and mechanical properties make it a fascinating material. As a renewable and biodegradable material, SCOBY aligns with sustainable design principles, allowing us to imagine alternative environmentally friendly solutions, reduce waste, and create products with a reduced environmental footprint. SCOBY can also be used on different scales, and as a substrate for bio-manufacturing.

Its various qualities make it suitable for different applications, such as healthy dietary fibers with prebiotic and antioxidant potential [36–38]. It can also be used as a functional ingredient like a thickener, stabilizer, texture modifier, for structuring, or as a pigmenting agent. It is widely consumed as a dessert in the Philippines; however, thanks to its versatile material states, it can be produced in various shapes and textures, such as films,

filaments, gels, powders, spheres, or solid foams. SCOBY can enhance the properties of products such as ice cream and yogurt, substitute animal derivates such as sausage casings, and potentially foster the development of novel foods such as snacks and fiber-based alternatives to standard carbohydrate products such as pasta. In the non-edible direction, SCOBY could be used for compostable packaging materials ranging from films, foam blocks, packing chips, or cosmetics. It can also be used as a functional ingredient for skincare and make-up, electronic components such as acoustic diaphragms and elements in compostable sensors with printed circuits, medical and pharmaceutical applications such as wound dressing, and for artificial blood vessels and scaffolds, thanks to its body compatibility; it enhances construction materials such as the mechanical properties of concrete, reducing cracking, and can be used for the bioremediation of water purification, and wastewater treatment [31,32]. The edible or non-edible suitability of SCOBY is highly dependent on the design of the liquid medium and the quality of the resources used. According to preliminary analyses conducted by food microbiology partners, SCOBY grown from tea appears to have valuable mechanical properties and less nutritional potential, making it a good candidate for compostable materials, while SCOBY grown from industrial apple secondary products seems to have lower mechanical properties yet prebiotic potential, making it more suitable for edible applications.

Various methods can be chosen to obtain SCOBY, from low-tech traditional static fermentation to high-tech bioreactors in a controlled environment. The main difference in selecting the method lies in the final destination of the SCOBY, in the access to resources and specialized equipment. Ideally, to obtain materials from SCOBY, high amounts of wet mass are needed for experimentation, requiring optimization and increased production. In this study, we present the development of the InnoCell bioreactor, a production unit that allows these goals to be achieved efficiently. The bioreactor presented in this study is part of the InnoCell research project, an interdisciplinary design-led project that seeks perspectives of circular glocal production. This research has been developed by the Design Friction Lab design team, resulting in open-source knowledge and tools that can be replicated, scaled, or improved, allowing designers and scientists worldwide to obtain SCOBY in large quantities and experiment with biomaterials. It is our hope that this development will foster the search for applications that support a circular economy, in this case, through the biological principles of fermentation.

To obtain SCOBY in reasonably high quantities and in a short period of time, a bioreactor is needed, which is a controlled environment artifact designed for the cultivation of microorganisms. There are several types of bioreactors for Kombucha fermentation used in different settings (Figure 2). Bioreactor choice depends on scale, process-control requirements, available resources, and the final destination of the SCOBY. Some common types of bioreactors used in the production of SCOBY are as follows:

Stirred tank bioreactors (a): These are commonly used in fermentations on an industrial scale. They consist of a tank equipped with an agitator for mixing and aeration. Stirred tank bioreactors allow for controlled fermentation conditions such as temperature, pH, and dissolved oxygen levels [39,40].

Airborne bioreactors (b): These use air or gas flow to create circulation and mixing within the reactor. They usually have a draft tube that serves as an air elevator and a descendant. Airborne bioreactors can provide efficient mixing and aeration, making them suitable for SCOBY production [39,41].

Packed bed bioreactors (c): These consist of a column filled with a solid support material, such as cellulose fibers or sponge-like matrices. The SCOBY grows on the surface of the support material, allowing for efficient nutrient utilization and gas exchange. Packed bed bioreactors can provide a large surface area for SCOBY growth, and are often used for continuous fermentation [39,42,43].

Tray bioreactors (d): This bioreactor type involves growing SCOBY in trays or plates. The trays are stacked vertically, allowing a large surface area for SCOBY growth. This type

of bioreactor is commonly used in home or small-scale environments, where simplicity and ease of handling are essential [43].

Rotating disc film bioreactor (e): These bioreactors serve to produce a cohesive film of cellulose. They consist of a series of rotating discs designed to allow microbial growth characterized by a high water ratio per unit weight of dry cellulose, compared to the cellulose produced under static conditions [39,44].

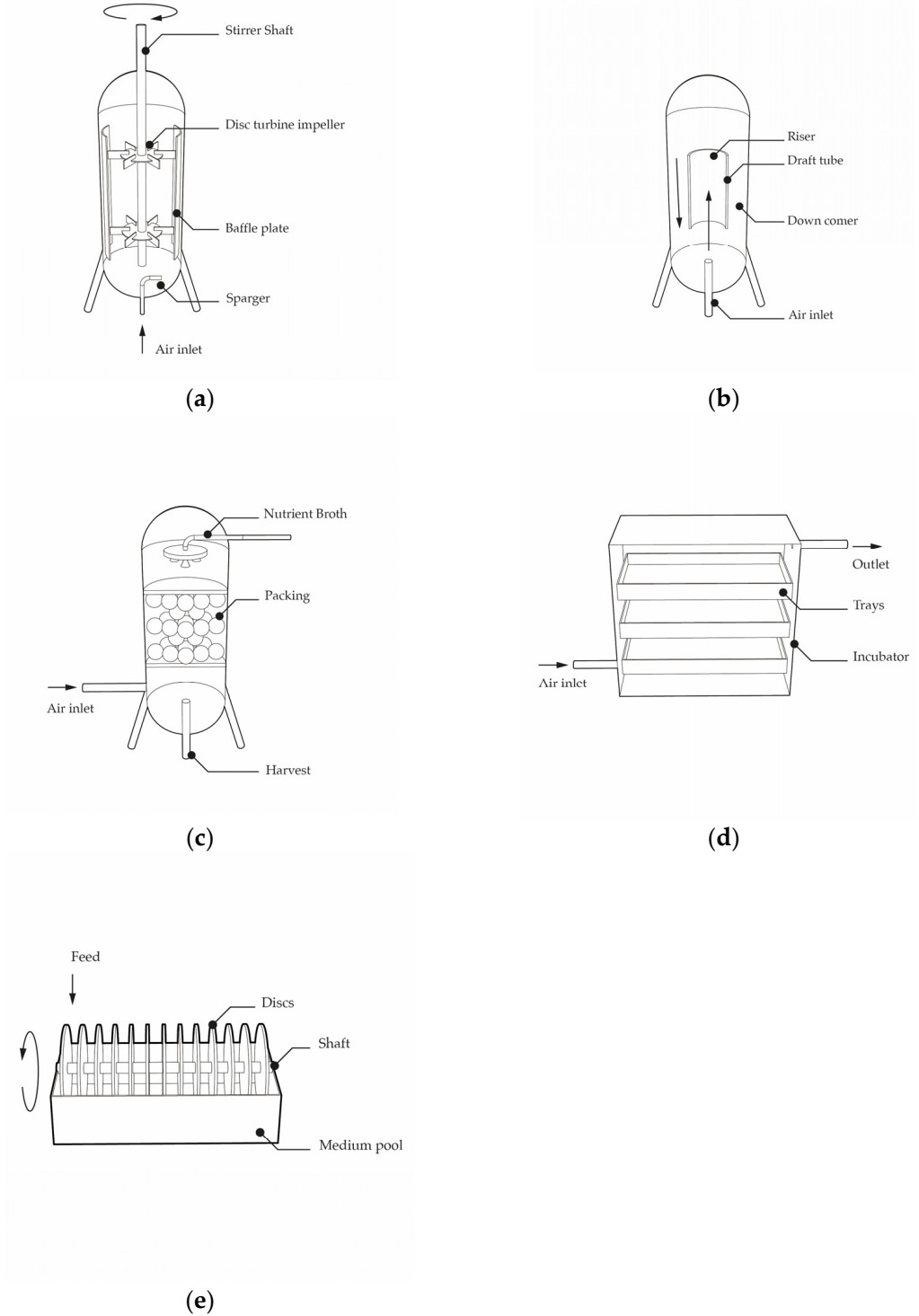

**Figure 2.** Common types of bioreactors for the production of SCOBY through fermentation. A SCOBY layer biofilm within them is generated from a fermentation cycle: (**a**) stirred tank bioreactor; (**b**) airborne bioreactor; (**c**) packed bed bioreactor; (**d**) tray bioreactor; (**e**) rotating disc film bioreactor.

As the bioreactor choice depends on the specific requirements of the SCOBY's production process, it is crucial to ensure that the chosen bioreactor provides optimal conditions for SCOBY growth, and maintains the parameters required over the production cycle for successful fermentation. Many bioreactors available in the market present a great technological complexity that not only increases the necessary investment, but also requires the intervention of engineering teams for assembly, production, and maintenance.

## 2. Materials and Methods

For the InnoCell project, continuous production of SCOBY was needed to obtain a sufficient quantity of material to test without compromising the time and scope of the research. For this reason, a rotating disc bioreactor was chosen. Different scholars recommend this type of bioreactor in terms of efficiency [39,44–46]; Bungay and Serafica developed in the early 1990s a bioreactor based on the rotating principle that allows for a high yield of SCOBY. This bioreactor's high production rate is excellent because it combines various factors that positively affect cellulose production [47]. Although the development was patented but has now expired (US5955326A), this patent served the design team to research and develop an open-source bioreactor under DIY principles that support the democratization of knowledge, a fundamental objective of the InnoCell project.

### 2.1. InnoCell Bioreactor Development

The InnoCell Bioreactor system consists of a rotating shaft that holds a series of perforated discs that serve as gripping elements for the microorganisms that secrete the microfibers, which thicken into pellicles. The shaft is partially immersed in a tank containing the liquid medium. Optimal oxygenation is achieved thanks to a constant and smooth orbital movement operated by a geared motor. Between 14 and 21 days, the SCOBY thickens on both sides of the discs, providing a noticeably higher yield (+95%) compared to static culture and agitated culture methods (+31%) thanks to surface optimization [45,46]. This fermentation duration difference compared to the beverage, circa two weeks, is due to the end product. While the liquid requires less time to achieve appreciable organoleptic qualities (7–14 days), the SCOBY requires more time to grow in thickness (14–21 days); in the extra time, the liquid becomes more acidic, fostering cellulose growth yet losing the possibility to be used as a beverage.

One aspect to consider is that the SCOBY produced on the disks has high water retention; therefore, we expressed the values in terms of wet mass instead of dry mass. The production of our final version spanned between 10 and 15 kg.

Another advantage is that the pellicles do not grow directly on the liquid but on the discs, leaving the liquid medium easy to access, enabling more practical monitoring and adjustments of the temperature, acidity (pH), and sugar content (°Brix). In addition, the rotating disc configuration considerably reduces the horizontal space needed for fermentation, opening new possibilities for industrialization and more space-efficient and stackable high-volume production.

### 2.2. Initial Prototypes

A preliminary experiment aimed to test if a rotating disc culture (RDC) could effectively provide greater yield than a static culture method. Two bioreactor prototypes were tested (Table 1). On one side, a bioreactor was used with rotating discs in two sizes on a polycarbonate GN 1/1 tank 530 × 325 × 200 mm with ø20 mm PVC tube as rotating shaft (Figure 3a). The discs had different hole patterns and were sanded, serving as gripping elements for the microorganisms. On the other side, a static culture used the same polycarbonate GN 1/1 tank 530 × 325 × 200 mm (Figure 3b). Both tanks were filled with the same amount of nourishing liquid, and were inoculated with SCOBY and previously fermented liquid (tea). Both prototypes had a fixed cloth to avoid contamination.

**Table 1.** First experiment testing two types of bioreactors.

|  | **Rotating Disk Culture Bioreactor (RDC)** | **Static Culture Tank (SCT)** |
|---|---|---|
| Culture ingredients | 12 L Tea.<br>4 L Fermented liquid (25%).<br>500 g of Sugar—Sucrose (3 °Brix).<br>185 g of Kombucha starter (SCOBY and liquid). | 12 L Tea.<br>4 L Fermented liquid (25%).<br>500 g of Sugar—sucrose (3° Brix).<br>185 g of Kombucha starter (SCOBY and liquid). |
| Rotating disk system elements | 11 disks laser-engraved and sanded with different grain patterns.<br>1 DC geared motor with encoder.<br>1 PVC tube.<br>2 MDF holders.<br>2 plastic bearings with inox spheres.<br>1 holder for the motor and speed regulation controller.<br>1 cloth-holding structure.<br>1 synthetic breathable cloth. | 4 plastic clamps.<br>1 cloth-holding structure.<br>1 synthetic breathable cloth. |

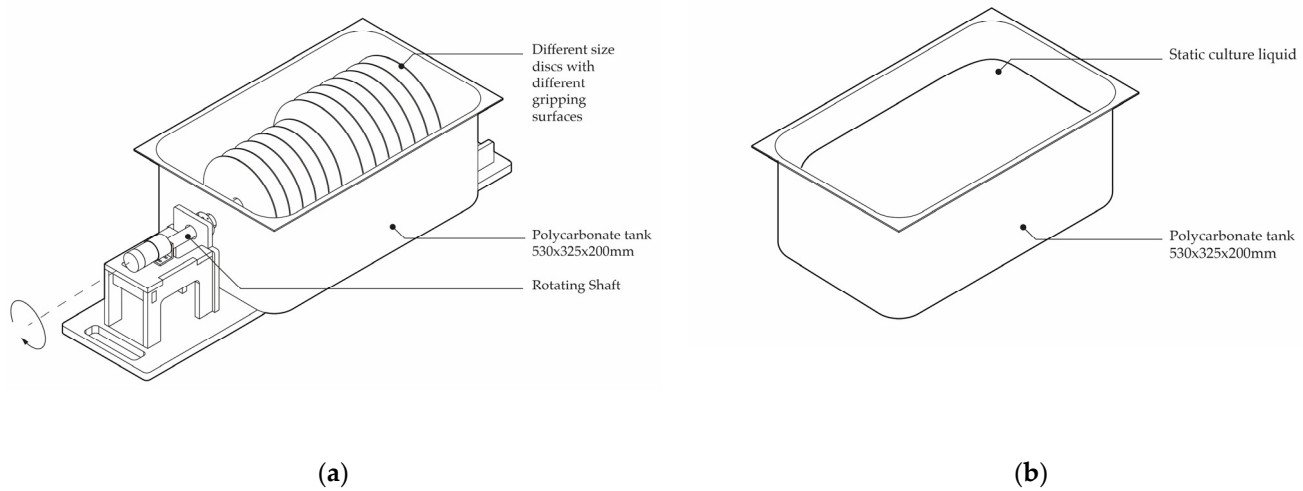

(**a**)  (**b**)

**Figure 3.** Initial prototype comparison test: (**a**) 16 L rotating disc culture bioreactor (RDR); (**b**) 16 L static culture tank (SCT).

A comparative experiment was prepared with the two culture prototypes (Table 2); however, the static culture tank was affected by contamination, presumably from spores and dust present in the air and the still-underdeveloped covering system. This was evident after the unpleasant smell and black mold spots that appeared on the surface of the liquid medium on day 5. The SCOBY growth was irreparably compromised, invalidating any effective comparison possibility. However, the SCOBY proved to grow efficiently on the different discs. The hole pattern presenting more wet mass was selected and implemented in the following iterations.

### 2.3. From the RDC Prototype to InnoCell Bioreactor Development

The InnoCell Bioreactor was developed to achieve enhanced SCOBY production and be easily reproduced with distributed technology available in workshops and Fablabs. After several tests, we identified a model of a geared motor with an encoder (12 V/28 RPM/80 kg) and a speed controller (5–30 V 6 A 150 W) as ideal for achieving the mechanical parameters

for SCOBY growth. The Food Technology Platform, our project partners, iteratively tested the liquid medium preparation protocol and fermentation control techniques. The final rotating culture bioreactor version produced 10–15 kg of wet SCOBY mass per cycle (Figure 4).

**Table 2.** pH, general composition of the starting medium, and growth process observations.

| Fermentation Time | Rotating Disk Culture Bioreactor (RDC) | Static Culture Tank Bioreactor (SCT) |
|---|---|---|
| Day 1 | Assembly. | Assembly. |
| Day 5 | Dots/little amounts of cellulose were seen attached to some disks. | Mold was spotted on the liquid medium. |
| Day 9 | Uniform layers of microbial cellulose grew homogeneously on some disks. | A thin molded layer of microbial cellulose grew slowly. |
| Day 14 | Uniform layers of microbial cellulose grew homogeneously on some disks. -pH of the liquid: Bottom left angle—pH 2.91 Bottom right angle—pH 2.85 Top right angle—pH 2.80 Top left angle—pH 2.83 Average pH: 2.85 | A thin molded layer of microbial cellulose grew slowly. -pH of the liquid: Bottom left angle—pH 3.10 Bottom right angle—pH 3.09 Top right angle—pH 3.07 Top left angle—pH 3.11 Average pH: 3.09 |
| Day 16 | Adjustment of the pH from 2.85 to 4.52 with a solution of water and circa 100 g of bicarbonate [1]. | A thin molded layer of microbial cellulose continued to grow slowly. |
| Day 21 | Microbial cellulose successfully grew on some disks. The system was disassembled, and the most efficient disk was weighed: the disk had a gross weight of 195 g. | A thin molded layer of microbial cellulose continued to grow slowly. |

[1] pH was measured in the four corners of the tank. According to the measurements of the pH meter, liquid on the RDC appeared to be more acidic than the one on the SCT, which is right; however, correction of pH (back to circa 3.5) was necessary to continue the fermentation process.

Accessories such as a removable fabric cover and heating system were also designed to maintain a constant temperature and avoid contamination, optimizing the fermentation conditions.

The InnoCell Bioreactor design choices are meant to be strictly functional. The predominant materials are PMMA (polymethyl methacrylate) and PC (polycarbonate) due to microbial suitability, food-gradeability, and to simplify the monitoring of the cultivation and growth of the pellicles from outside the tank. The components and structures are designed to be easily disassembled for easy handling and cleaning. The half-cylinder shape of the tank optimizes the interior volume, following the profile of the twenty-eight discs with a diameter of 250 mm. The discs are mounted on a square profile polycarbonate shaft, separated with 3D-printed spacers and stoppers. A specific pattern of holes, together with a surface treatment (coarse sanding), optimizes the growth of the pellicles on the discs (Figure 5). The last performed cycle generated wet mass with an average of 500 g per disc, reaching a total yield of 13.5 kg.

The InnoCell Bioreactor mainly uses digital fabrication tools (3D printing, laser cutting) that makes the construction process more accessible, re-producible, and low-cost. Accessories like a removable cloth cover and a heating system (made of a filter, universal aquarium pump, vertical liquid heater, and a metal holder) are broadly available in the market. The bioreactor also requires a motor and its power supplier and additional components (tubes of varying diameters to connect the pump to the heater to the liquid medium).

The downloadable open-source instruction manual has detailed step-by-step guidance for reproducing the InnoCell Bioreactor (Supplementary Materials) [48].

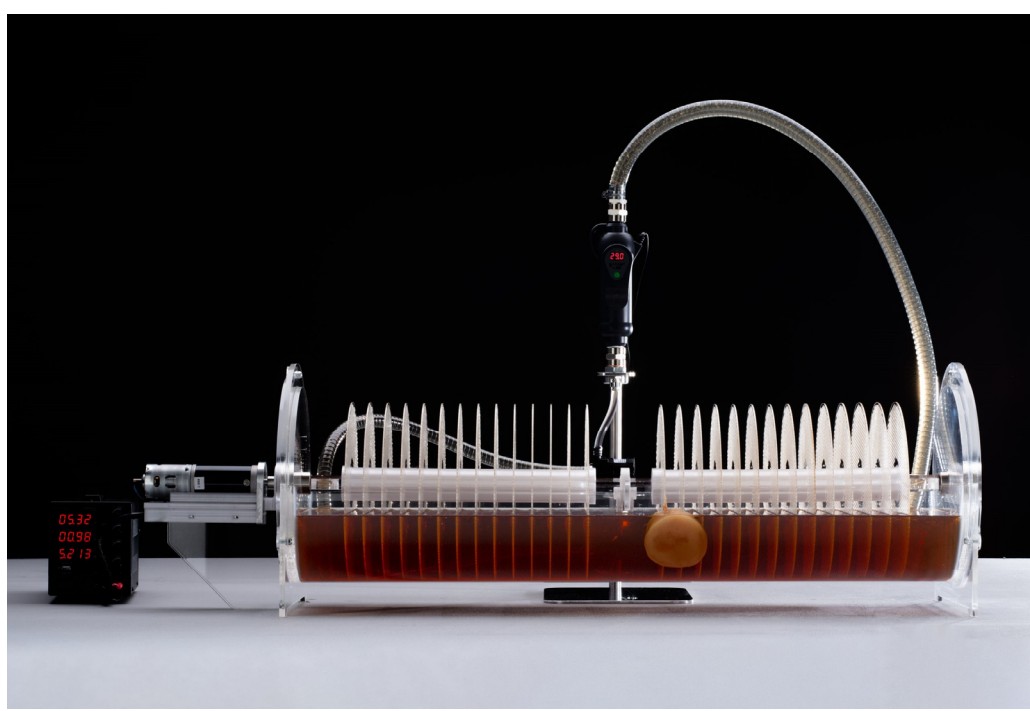

**Figure 4.** InnoCell Bioreactor with the liquid medium ready to begin a fermentation cycle.

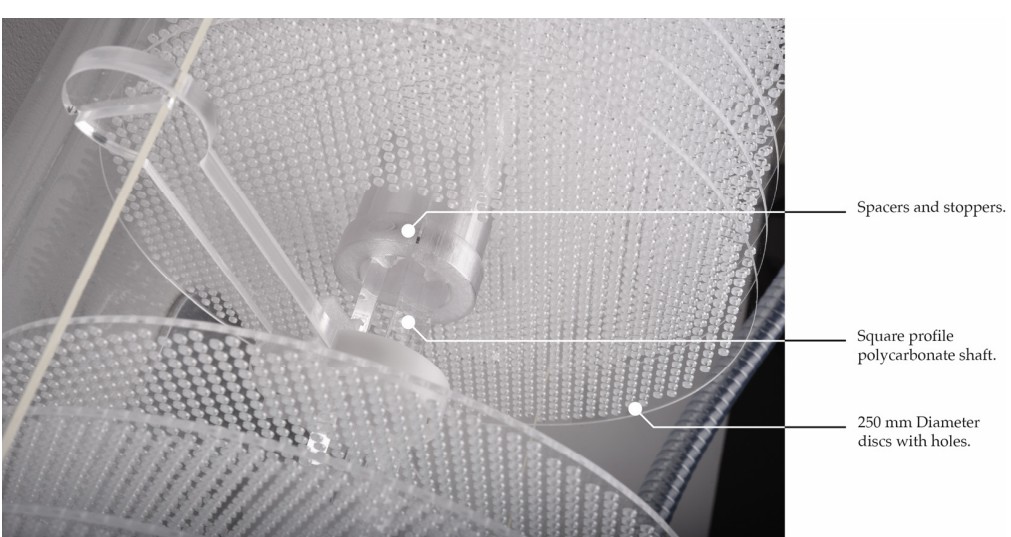

Spacers and stoppers.

Square profile
polycarbonate shaft.

250 mm Diameter
discs with holes.

**Figure 5.** InnoCell Bioreactor detailed components.

## 3. Results

The iterative practice-based development of the bioreactor conducted to meta-level knowledge production on designing is not only a tool, but also an approach that goes beyond being a mere medium for material production. The results highlight advantages of producing SCOBY with this artifact.

### 3.1. Producing SCOBY with the InnoCell Bioreactor

To start producing the SCOBY pellicles aimed to become fermentation-based biomaterials, some considerations must be taken:

The size of the bioreactor was developed to be a one-meter module (size of the fermentation tank), allowing simple handling and management (Figure 6). As such, the unit can be used to build multi-module systems for a larger scale production. The InnoCell Bioreactor has the following features:

- 25 L capacity (full tank);
- 28 disks (ideal growth per disk 350–600 g);
- 10–15 kg wet pellicle-mass production per cycle.

1. Tank shape: As the disks are round, it seemed logical to follow their shape for volume optimization. Indeed, a study proved that efficient pellicle growth depends more on the availability of nutrients and oxygen rather than liquid medium volume [49]. Moreover, in the angles of a square-profile tank, undesired elements can concentrate and more easily contaminate the culture. Optimizing the volume allows more cellulose mass with less liquid. The "u" shape (or half-cylindrical) of the bioreactor optimizes the quantity of liquid.

2. Shaft: The square profile of the shaft keeps the disks fixed, facilitating a more homogeneous pellicle growth, and prevents them from rolling against each other. This also avoids the discs from spinning loosely by the effect of friction and wear of the axis hole in the long term.

3. Motor/bearings/board holders: To minimize alignment issues and weight fatigue for the motor, the various components, discs, and spacers are divided in the middle after the 14th disc by a bearing mounted on a place holder. The motor is mounted on a holding structure that is external to the tank to minimize the contact with the liquid medium. The motor and the power bench should be boxed to minimize the risks of contact between liquid and electricity.

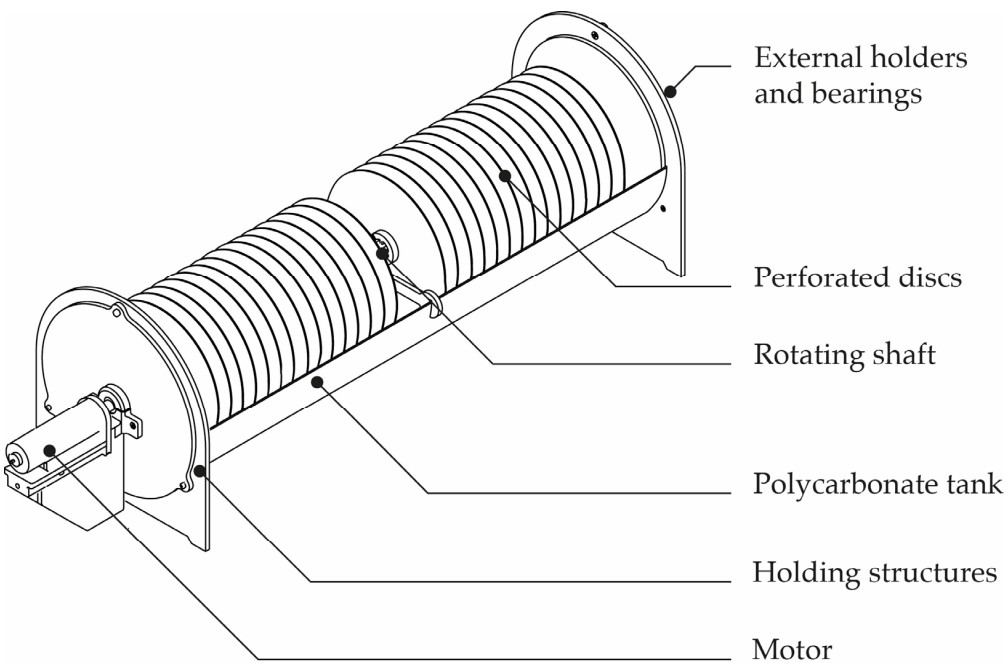

External holders and bearings

Perforated discs

Rotating shaft

Polycarbonate tank

Holding structures

Motor

**Figure 6.** Innocell bioreactor module.

*3.2. Considerations for an Effective Fermentation*

3.2.1. Capacity and Liquid Parameters

- The InnoCell Bioreactor presented in this study has a capacity of 25 L and yields, depending on conditions (nourishing medium, pH, temperature, time), from 10 to 15 kg over a fermentation time of 14 to 21 days. The liquid medium needs to be refilled on an ongoing basis to ensure growth of the cellulose on as much surface of the discs as possible; the lower the liquid level, the less surface of disc is in contact

with nourishment, and no thickening will occur. In our experience, circa 25–30 L of refill is needed throughout one cycle.

- It is recommended to have a starting liquid with a pH of 4–4.5 and a concentration of 5 °Brix degrees (50 g/L) that can be achieved with different sugar sources (with tea-based medium, circa 18.75 L of sweetened tea + 625 L of already fermented tea). If fruit and vegetable secondary products are used, it is important to measure the actual sugar content, as this may strongly influence the fermentation process. In this case, the biomasses chosen needs to be pre-treated to inactivate contaminants by boiling the mass for 30 min if necessary after dilution with water (e.g., in the case of apple pomace, 1:3 ratio—1 part of apple mass in 3 parts of water) or autoclaving at 121 °C for 30 min (heating treatment may, however, denature some valuable substance such as certain vitamins). It is important to start from an organic mass that is not rotten or heavily contaminated by fungi. Various companies already store the secondary products in the forms of dry pellets or concentrated packaged masses, which are more stable, and therefore suitable for this process. However, for such decisions, consultation with food technologists, microbiologists, and scientists is crucial. After the pre-treatment, the liquid needs to be filtered properly, as particles would be otherwise incorporated into the pellicles, decreasing their mechanical properties. Brix degrees were measured using a pocket refractometer (VWR Digital Handheld Refractometer Cat. No. 75997-572 0-54 Brix 1.33-1.42 RI. Manufactured by VWR International, LLC. Radnor, PA 19087 USA).

### 3.2.2. Speed

The motor is connected to the power bench and the speed needs to be set. The rotations per minute (RPM) should be adapted to the growing medium, also in relation to the growing pellicle. In our experience, the right speed for a tea-based medium was 11 RPM, while for apple pomace-based broth it was 8.5 RPM, because of different mechanical properties generated by the different media.

### 3.2.3. Acidity

- In our experience, the starting pH should be between 4 and 4.5. This is more acidic compared to reviews that report an ideal pH of 4 to 6. However, especially in non-controlled environments, starting from a lower pH significantly decreases the contamination risks from certain spores present in the air and, more importantly, from pathogens.
- Such contamination risks increase in warmer seasons due to the more favorable room temperature. In a case of ongoing contamination during the summer of 2020, we started from a pH of 3.8, as suggested by the Food Tech Platform, using fermented liquid and acetic or citric acids, in order to discourage in the early stage the development of contaminants. After the pellicles started to grow, we raised the pH to 4–4.3 through refills. However, for a proper control over spores, an air filtration system should be added.
- During the fermentation, the pH should not drop below 3.5, as this would slow down and even stop the pellicle production. Therefore, the pH should be monitored with a pH meter; when it approaches 3.5, it should be raised to 4–4.5. This works sometimes by solely refilling (in other words, via dilution); however, in case of dramatic acidity, a solution of water and bicarbonate can be used.

### 3.2.4. Refill

- During the fermentation, part of the liquid evaporates; therefore, the tank should be refilled and adjusted with medium or fermented liquid (which may possibly cause changes in the pH). The fermentation generates acids that protect the culture. Constant monitoring enables the grower to adjust the liquid parameters to ideal conditions. The liquid volume should always reach the level of the holes pattern in the disks (circa the halfway) and be controlled every second day. If the liquid volume is too low, the

pellicles may only grow in the external area of the disc and possibly cause drying of the left-over inner areas (Figure 7).

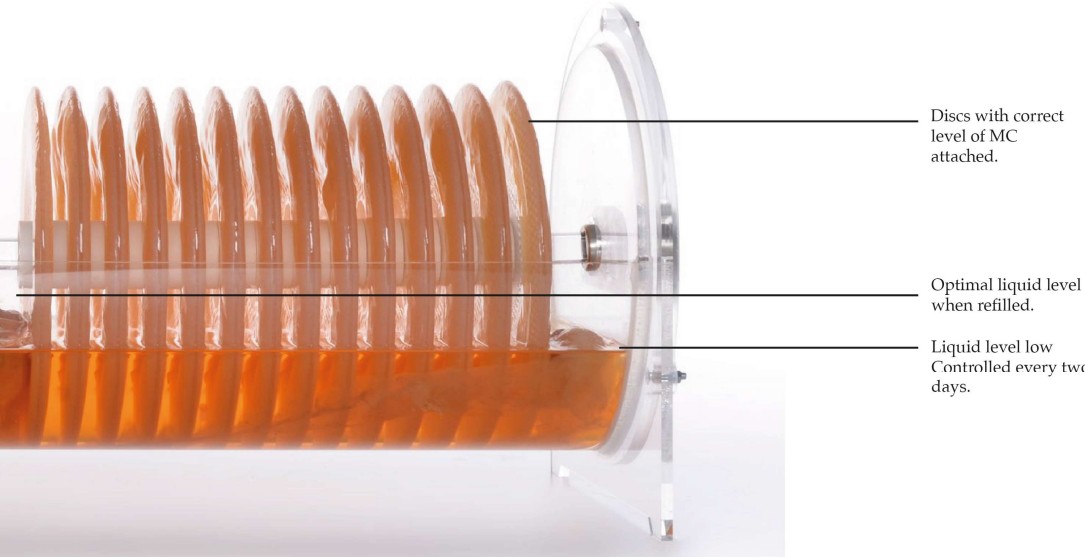

**Figure 7.** InnoCell Bioreactor during fermentation process.

3.2.5. Heating

- A constant temperature of 28–30 °C should be maintained to ensure efficient growth. This can be provided by an external heating system composed of an aquarium pump and vertical heater connected to the tank through flexible plastic tubes. The system provides constant circulation to ensure culture homogenization. The heating system creates a circulation of the liquid inside the tank, which maintains the homogeneity of the medium.

3.2.6. Cover

- A cover based on a breathable cloth and straps was made to protect the fermentation culture from air contamination, and to slow down heat dispersion and liquid evaporation. It was efficient to the scope of the InnoCell research, and could be further adapted and developed.

3.2.7. Collection

The pellicles can easily be collected by hand by literally peeling them off the disks (Figure 8). They can be stored in a wet state in a container (Figure 9). The pellicles are round, with a hole inside and carry the hole patterns in relief; therefore, they are not suitable for use in sheets. They are more suitable for use after homogenization (blending).

3.2.8. Cleaning

- The whole system should be disassembled and cleaned after the end of every cycle. It is crucial to remove the holding structures and the motor, putting them aside. The shaft with the disks should be completely disassembled and cleaned with a sponge and/or brush with dishwashing soap, hydrogen peroxide, or sodium hypochlorite. The same procedure should be carried out for the tank. The system cannot be autoclaved. The fabric used above the tank was washed at 90 °C in a common washing machine before every cycle.

Attention: Alcohol and ethanol could crack the PMMA disks; thus, avoid using these. Alternative polymers for the disks could be PE (polyethylene), PP (polypropylene), HDPE (high-density polyethylene), or PC (polycarbonate), which can all be food-grade and compatible with ethanol.

- The biofilm grows everywhere in the liquid medium or passes through; hence, building any heating circulation system would very likely allow it to grow inside and clog it. In order to avoid components being damaged, the system should be checked daily, and if the liquid flow slows down, the heating system should be disassembled and cleaned with the aid of a pipe cleaner and reassembled.
- In our experience, clogs may occur about 2–3 times per cycle. Plastic tubes with a large inner diameter discourage clog formation.

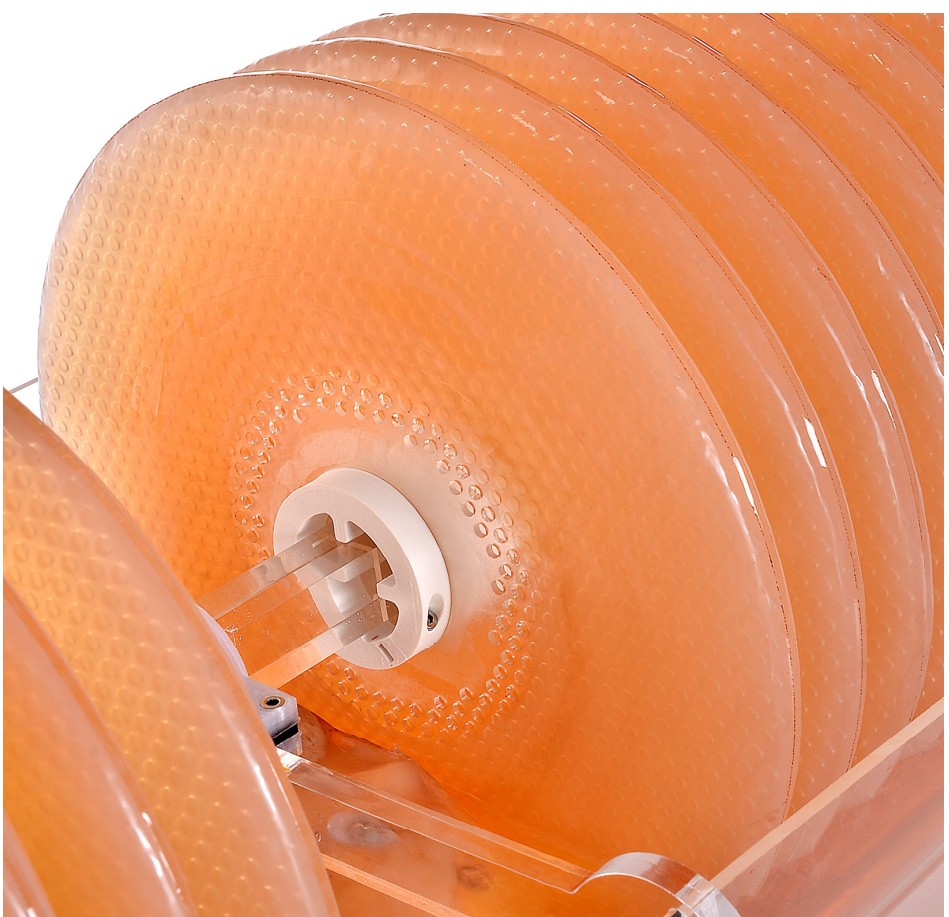

**Figure 8.** Microbial cellulose attached to the discs.

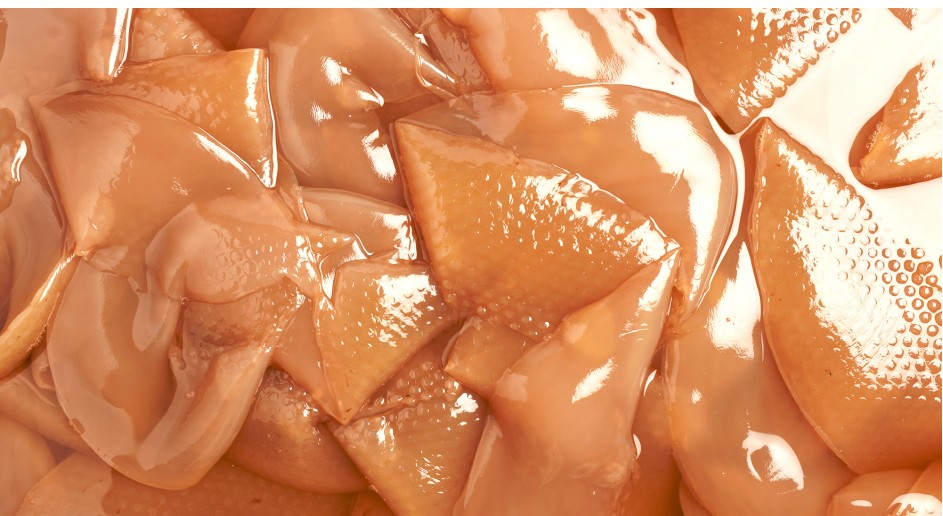

**Figure 9.** Harvested microbial cellulose collected after removal from discs.

*3.3. Fermentation Process with the InnoCell Bioreactor*

Instructions

1—Prepare the liquid following the guidelines (25 L of the specifically designed nourishing liquid according to raw resource's sugar content, e.g., apple mass: 1:3, included in the 25 L circa 5–6 L. 25%—should be already fermented liquid, as stated by the Food Tech Platform).

2—Clean the tank and the disks using hydrogen peroxide.

3—Pour the liquid into the tank (25 L).

4—Control the temperature: When below 30 °C, place SCOBY (literature suggests 3% of the volume [50], but we noticed that also 1.5–2% is adequate) into the liquid.

5—Control the power supply, ensure it is set at 12 V, and turn on the power strip.

6—Set the direction of rotation.

7—Set the rpm to 11 if the liquid medium is tea; the motor will start turning and will remain so until the end of the fermentation cycle (14–21 days).

8—Control the central bearing to be aligned with its holder.

9—Cover the system with a protective cloth.

10—Let the fermentation take place (Figure 10).

11—Check the system and its values daily (pH-|BRIX-| heating system). Adjust when needed, clean clogs when they occur.

12—Collect pellicles of MC.

13—Disassemble and clean.

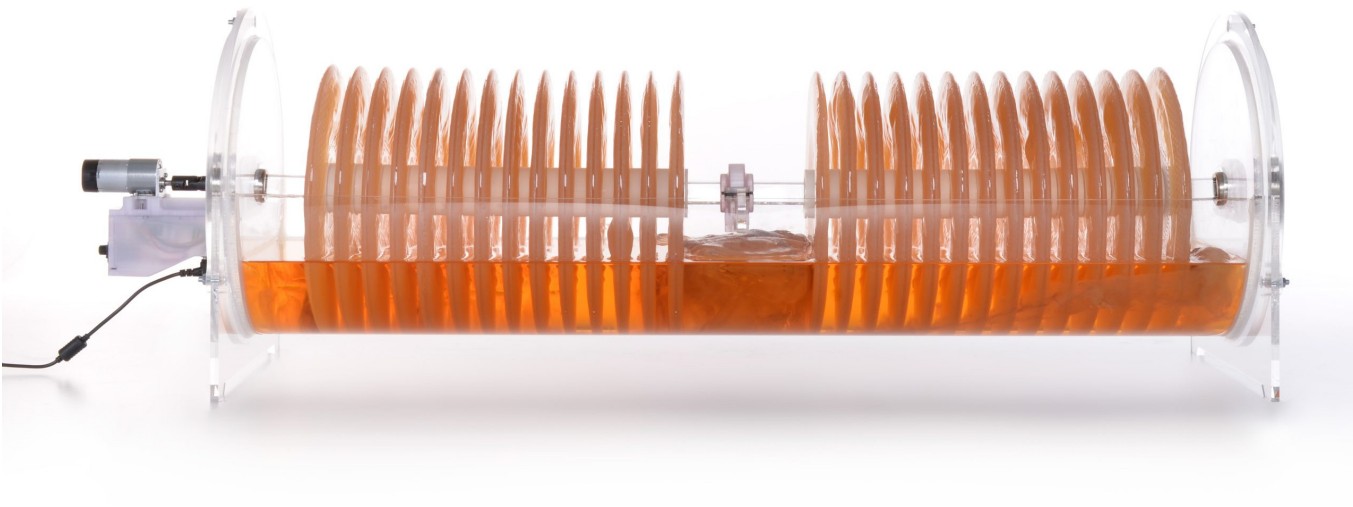

**Figure 10.** The InnoCell Bioreactor at the end of a fermentation cycle.

## 4. Discussion

In this study, we wanted to illustrate the iterative process carried out on an optimized unit for SCOBY production, and how a biomaterial may be grown using a rotating disc bioreactor to enhance a fermentation process. By comparing different types of production techniques and understanding the possibilities a rotating disc culture offers, we also wanted to draw attention to the valuable results that multidisciplinary knowledge can exchange between design and science. The link between the extensive knowledge, which scientists possess on the fermentation processes, combined with the knowledge of designers in the making and iterative learning by doing, produced a machine that can easily be manufactured and used to produce a broad spectrum of biological materials to tackle the current crisis of production and consumption in multiple areas around the globe.

We know that people experience materials in different forms, and the urgency to propose alternatives for edible (food grade) and non-edible applications becomes necessary in a society that needs to shift towards sustainability and circularity. We do not suggest that

using a bioreactor will immediately produce feasible and scalable solutions. Still, the more access that is given to such artifacts to a growing experimental community of scientists and creatives, the higher the chances are to implement new sustainable biological materials into our production and consumption systems. A straight study of how to transform relatively complex equipment into an open-source accessible machine would not necessarily drive ready-to-market materials. The InnoCell research provided an opportunity to gain experience on the potential of Kombucha fermentation and SCOBY materials grown from apple pomace-based nourishing liquids. However, extensive studies on characterization and consolidation need to be performed.

As pointed out in [51], the current state of the art on methods for continuous production of SCOBY in sufficient quantities that optimize the time and scope of development of a growing material has been an unexplored topic in the emergent bio-design research field. This study contributed to this direction by providing a remarkable tool that is easy to build and use by communities and emerging practice-oriented teams willing to grow materials.

The InnoCell bioreactor is a fundamental part of the InnoCell project that aimed at creating alternative material solutions that can be edible and/or non-edible for sustainable futures. The developed bioreactor allows R&D acceleration towards integrated applications. We hope other labs can also benefit from constructing InnoCell bioreactors and share their outcomes in a more open and shared community for the circular economy transition [52].

## 5. Conclusions

In conclusion, this study introduced a significant advancement in the production of versatile biological matter by developing an open-source bioreactor designed for SCOBY-induced bacterial cellulose production. This innovation underscores the benefits of the rotating disk culture method, representing a further advancement in efficient bacterial cellulose generation. By emphasizing material development from a design perspective, the interdisciplinary team took on the challenge of transforming complex laboratory tools to produce an understandable yet robust device, incorporating a valuable shift from the laboratory to society [53].

With this intention, it hopes to inspire and support other teams to engage in material fermentation, and to enhance their own experimental endeavors.

Crucially, this collaborative effort between design and science, rooted within open-source sharing, emerges as a tool for propelling research towards new materialities for the transition to an effective circular economy. This aligns with the broader goals of social sustainability; wherein open knowledge democratizes access to sustainable development.

It is imperative to balance this open sharing of knowledge and the protection of creative freedoms, exemplified by our embrace of Creative Commons licensing. While the bioreactor presents significant advantages, it has its challenges. Variations in liquid medium composition and culture conditions influence SCOBY growth, its bio-mechanical properties, and morphological characteristics dictating the pellicles' suitability for more specific uses. Additionally, determining maximum thickness and morphology remain areas of ongoing investigation.

**Supplementary Materials:** The Innocell Bioreactor Production Manual can be downloaded at: https://designfrictionlab.com/project/bioreactor/ (accessed on 6 September 2023).

**Author Contributions:** Conceptualization, N.C.; methodology, S.U.Y.; investigation, E.S. and I.M.S.-F.; resources, E.S.; data curation, L.C.; writing—original draft preparation, C.A.-G.; writing—review and editing, C.A.-G., N.C. and E.S.; visualization, E.S. and C.A.-G.; supervision, N.C.; funding acquisition, N.C. All authors have read and agreed to the published version of the manuscript.

**Funding:** This research was funded by the Free University of Bozen-Bolzano, internal call and grant number: unibz-id-2017. This work was supported by the Open Access Publishing Fund of the Free University of Bozen-Bolzano.

**Institutional Review Board Statement:** Not applicable.

**Informed Consent Statement:** Not applicable.

**Data Availability Statement:** The data presented in this study are available in https://designfrictionlab.com/project/bioreactor/ (accessed on 6 September 2023).

**Acknowledgments:** Design Friction Lab: Matteo Scalabrini; Food Technology Platform: Matteo Mario Scampicchio, Giovanna Ferrentino, and Haman Nabil; BITZ Fablab: Uwe Felderer and Kathrin Kofler; Faculty of Design and Art Workshops; Roland Verber, Valentin Riegler, Albert Kofler, and Robert Mößler.

**Conflicts of Interest:** The authors declare no conflict of interest.

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
