# Peer review of "Innocell Bioreactor: An Open-Source Development to Produce Biomaterials for Food and Packaging Based on Fermentation Processes"

_fermentation, doi:10.3390/fermentation9100915_

Round 1

Reviewer 1 Report

1. Please provide more investigation on the contamination during the comparative study between the 2 prototypes, as decontimination will be quite important for large scale production. 

2. Please provide legends on the Innocell bioreactor. In your manuscript, the pictures of the bioreactor do not show reviewer enough information, e.g. how the samples are taken, how the pH is controlled, how do you re-fill the medium during culture without introducing contamination, etc...

3. Please consider other materials of construction for the disc as it is not compatible to ethanol. The ethanol is a comment disinfectant in industry. 

Excellent wording and clear expression of the study. 

Reviewer 2 Report

The authors presented interesting research results, but some aspects related to methodology should be extended.

The digital format of the full text needs to be uniform.

The reference format of the article is not uniform. Please pay attention to the citation format.

Reviewer 3 Report

Overall, the manuscript "Innocell Bioreactor: An open-source development to produce biomaterials for food and packaging based on fermentation processes" reports the development of a novel bioreactor through the use of more accessible, reproducible, and low-cost fabrication tools to produce new sustainable material, particularly for food and food materials packaging applications. In addition, the opportunity for readers to have access to the instruction manual with details to reproduce the InnoCell Bioreactor is really interesting.

The manuscript is well written, organized, and easy to read, and the concept is suitable to publish in Fermentation. Therefore, the same is suggested for publication after these minor revisions:

1- Please give the full name for K. hansenii (Page 3, line 107-108).

2- Please enumerate what are more specifically the edible and non-edible potential applications for the obtained materials from SCOBY, and clarify why these materials are more suitable for food and packaging applications.

3- The fermentation process conducted for the Kombucha beverage usually takes between 7 and 14 days to produce the SCOBY (Pages 2-3, lines 96-100), while the InnoCell Bioreactor takes about 14 to 21 days (Page 11, lines 340-342). Please, can you explain this reason?

4- You should explain in more detail how the composition and the origin of the liquid medium can influence the SCOBY growth.

5- Are the microbial cellulose pellicles formed on the different discs similar in terms of thickness and morphology? What is the maximum thickness that the pellicles can obtain on the discs?

Reviewer 4 Report

Comments for Authors: 

This is a well written document that presents the process of “optimizing a bioreactor unit for SCOBY production”. Though the document is clean, with minor errors, and the aim of the publication might fit the MDPI Topics (Bioreactors: Control, applications and optimization), this reviewer considers it does not have the appropriate format for the Fermentation Journal. The Introduction and Materials and Methods sections are accompanied with interesting theoretical aspects that might give the reader the idea that it is written as a Book Chapter, rather than as a research paper; the content is very informative and useful, full of interesting and detailed information.

As a reviewer, I consider this is a high-quality manuscript, amenable for readers, full of important and motivating content, especially for those involved in the field of Bioreactor design and operation, under the DIY principles.   However, extra theory must be either remove or at least reduced, for letting the prototype shine as the main outcome of the work. Other sections need also attention, for instance, the Conclusions section does not conclude; it rather summarizes the content of the manuscript.

This reviewer humbly suggests authors must adopt the format/guidelines the journal offers (for authors).
